# Normative Data on Grip Strength in a Population-Based Study with Adjusting Confounding Factors: Sixth Korea National Health and Nutrition Examination Survey (2014–2015)

**DOI:** 10.3390/ijerph16122235

**Published:** 2019-06-25

**Authors:** Seong Hoon Lim, Yeo Hyung Kim, Jung Soo Lee

**Affiliations:** 1Department of Rehabilitation Medicine, St. Vincent’s Hospital, College of Medicine, The Catholic University of Korea, Seoul 06591, Korea; seonghoon@catholic.ac.kr; 2Department of Rehabilitation Medicine, Uijeongbu St. Mary’s Hospital, College of Medicine, The Catholic University of Korea, Seoul 06591, Korea; kyhyung@gmail.com

**Keywords:** hand strength, dynamometer, reference value

## Abstract

*Background:* We investigated population-based data on grip strength, analyzed for demographic factors, and proposed a formula to estimate grip strength that could be generalized to a population with different anthropometric and background characteristics. *Methods*: This study used a complex, stratified, multistage probability cluster survey with a representative sample of the population. Select household Korean participants (*n* = 6577) over age 10 who were able to perform daily tasks without issue were included. Grip strength was measured in both hands, alternately, three times using a digital grip strength dynamometer. *Results*: There was a curvilinear relationship between grip strength and age, and grip strength was higher in males than females (*p* = 0.001). Hand preference significantly affected grip strength (*p* = 0.001). Weight and height were positively correlated with strength in both hands (*p* = 0.001), but waist circumference was negatively correlated with strength in both hands (*p* = 0.001). The intensity of occupational labor did significantly affect grip strength in both hands (*p* = 0.001). The formulas for estimating grip strength of each hand are presented as main results. *Conclusions:* To determine normative data on grip strength, we may consider factors such as occupations with different physical demands, underlying medical conditions, anthropometric characteristics, and unmodifiable factors such as age and sex.

## 1. Introduction

Grip strength is a simple tool used to rapidly assess muscle strength [1,2]. It can provide diagnostic information and be used to assess clinical outcomes [3,4,5]. It has been used as a predictor of overall body strength, function, survival outcome in disease, and nutritional status in old age [1,6,7,8]. Measures of grip strength are important in geriatric and occupational assessments. However, it is influenced by various factors such as age, sex, weight, and height. Age and sex are major factors influencing grip strength in young and old people, and previous studies have demonstrated that occupation and cognition also influence grip strength in old age [9,10]. Therefore, the normative reference value of grip strength may be difficult to estimate without considering multiple contributing factors, even when using data with sufficient sample size. In addition, predicting grip strength with formula may have merit for various clinical assessments.

In this study, we investigated normative data for grip strength by age and assessed potential factors associated with grip strength in a large population of Koreans aged at least 10 years old. We suggest a formula model for estimating grip strength using different anthropometric and background characteristics.

## 2. Material and Methods

### 2.1. Study Design

This study was based on data of the sixth Korean National Health and Nutrition Examination Survey (KNHANES VI), 2014–2015. This was a transversal national health examination survey that collected a representative description of the state of health, nutritional status, and physical activities of the general Korean population. To recruit a representative sample of the noninstitutionalized civilian population aged 1 year and over in Korea, the whole nation was stratified into 16 provisional regions of non-apartment residents and apartment residents. Then it was stratified again by cluster analyses using the population rate of age groups, under 14 year old, over 66 years old, and per capita living space within each first stratum, using the enumeration district of the 2010 Census as a sampling frame. Each survey year includes a new, different sample of about 10,000 individuals aged 1 year and over obtained by systematic sampling. All data are presented as an estimated mean ± standard error with 95% confidence interval (95% CI).

KNHANES was conducted by the Division of Chronic Disease Surveillance, Korean Centers for Disease Control and Prevention (KCDC). The survey collected data via household interviews. Standardized physical examinations and assessments were performed by well-trained examiners in specially equipped mobile examination centers, after obtaining informed consent from all individual participants. Detailed information about the survey is available on the KNHANES website (https://knhanes.cdc.go.kr). Informed consents to participate in the study were obtained from all participants by the KCDC [11]. The study was exempt from approval by the Institutional Review Board of Catholic University, College of Medicine, as the study utilized a publicly accessible database, covered by the KCDC.

### 2.2. Participants

This study included children ≥ 10 years old and adult participants who could fully understand and cooperate during interviews and assessments. Each survey took from 1 h 30 min to 2 h to complete. In the survey, health is defined by the presence or absence of disease. However, only medical staff can judge health through examination, and in middle-aged and elderly people, it is not appropriate to define health as merely the presence or absence of disease, because a person who declares to healthy may have a chronic disease. Thus, we defined healthy participants using the EuroQol 5-Dimension Questionnaire (EQ-5D), a self-reported questionnaire about daily activities and social activities, which is a standardized measure of health status developed by the EuroQol Group to provide a simple, generic measure of health for clinical and economic appraisal [12]. The EQ-5D measured health perceptions and consist of a descriptive system with five dimensions (mobility, self-care, usual activities, pain/discomfort, anxiety/depression). The EQ-5D is a useful tool for assessing health-related quality of life (HR-QoL) [13,14]. Participants are asked about their current health status (with the following answers: 1 = no problem; 2 = some problems; and 3 = severe problems). The EQ-5D index is calculated using estimated weighted quality values for Koreans, with scores ranging from 1, representing perfect health status, to −1, representing a health status that is no better than death [15].

A total of 6577 participants, who reported no problems in all 5 dimensions and responded “no” to the question “Have you recently had any problems in daily activities and social activities due to underlying medical conditions or physical or mental disabilities?” were ultimately included.

Detailed standardized protocols of all covariate measurements and a guidebook about the quality control of measurements are available on the KNHANES website (https://knhanes.cdc.go.kr).

### 2.3. Grip Strength Measurement

Respondents with amputations, limb deformities, fractures, paralysis, or casting/bandages on the arm, hand, or fingers, including the thumb, were excluded from the grip strength measurements. Those who had had hand or wrist surgery within the prior 3 months or who had experienced any pain, tingling, stiffness, discomfort, or aggravation in the hand or around the wrist within the prior 7 days were also excluded.

Maximum grip strength (kg) was measured in both hands, three times, alternately, using a digital grip strength dynamometer (Takei Digital Grip Strength Dynamometer Model TKK5401, TAKEI, Niigata, Japan), which is sensitive to small changes that can serve as indicators of the clinical picture of disease [16,17,18].

Grip strength was measured by four different nurses who were regularly trained by quality managers. Quality managers also inspected whether grip strength measurements had been measured according to the standardized protocol. The participants looked forward while standing upright with shoulders straight and both arms falling naturally to either side. The participants did not bend their elbows or wrists and their arms did not touch the body or take an attention posture. Feet were positioned under the hips with toes pointing forward. The participants maintained this basic posture while grip strength was measured. The examiner first measured the hand participants preferred to use in daily activities and gave a 60 s rest time after measuring both hands. It was measured three times in the same manner. The detailed guideline of grip measurement is available at KNHANES website (https://knhanes.cdc.go.kr/knhanes/sub04/sub04_02_02.do?classType=4).

The maximum grip strength was taken for 6168 right hands (93.78%) and 6184 left hands (94.02%) from a total of 6577 participants. Intraclass correlation coefficients (ICC) with 95% CIs were used to assess relative reliability across grip measurements. The ICC (95% CI) and CV across the three repeated grip measurements were 0.985 (95% CI: 0.984–0.985) and 34.8% in right-hand grip measurements and 0.988 (95% CI: 0.987–0.988) and 35.3% in left-hand grip measurements. The mean maximum grip strength from the three measurements was used for statistical analyses.

### 2.4. Covariates

Anthropometric and demographic covariates included chronological age, sex, body weight, height, waist circumference, hand preference, and current occupation. Body weight (unit: kg Seca 225, Seca, Hamburg, Germany), height (unit: cm, GL-6000-20, G-tech, Seoul, Korea), and waist circumference (unit: cm, Seca 200, Seca, Hamburg, Germany) were measured by well-trained public healthcare staff who were regularly trained by quality managers. Quality managers also confirmed that measurements had been taken according to a standardized protocol. Hand preference was obtained using the following question: “Which hand do you use to perform most tasks?”. Current occupation was classified into 11 categories according to the official occupation classification of Statistics Korea: (1) managers, (2) professionals and related workers, (3) clerks, (4) service workers, (5) sales workers, (6) skilled agricultural, forestry and fishery workers, (7) craft and related trades workers, (8) equipment, machine operating and assembling workers, (9) elementary workers, (10) armed forces, (11) student or housewife or no occupation. We reclassified occupations with different physical demands into three categories: high (5–10), medium (1–4), and low (11).

Underlying medical conditions with pain symptoms, including diabetes mellitus (DM), hypertension (HBP), and knee osteoarthritis (OA), were included as covariates, because these medical comorbidities are commonly encountered in an older population, although participants responded that they had no problems in daily and social activities [11,19]. DM was defined as fasting glucose levels (in serum) of ≥126 mg/dL, use of anti-diabetic medications, or a physician’s diagnosis of DM. Impaired fasting glucose was defined as fasting glucose levels (in plasma) of 100–125 mg/dL. Fasting glucose levels (in the blood) were measured using an enzymatic method. HBP was defined as a systolic blood pressure ≥ 140 mmHg, a diastolic blood pressure ≥ 90 mmHg, or use of anti-hypertensive medications, and participants were dichotomized by HBP status [19]. Blood pressure was measured by trained medical staff following a standardized procedure during the health examination survey [19]. Knee osteoarthritis was defined by a physician’s diagnosis and/or a patient taking medications for this condition at the time of self-reporting [11]. Underlying medical conditions were classified into two groups according to a “yes or no” questionnaire.

### 2.5. Data Analysis

The characteristics of participants were analyzed via descriptive methods for complex samples and stratified according to sex and age level. The normative data on grip strength were analyzed and estimated with the complex-samples general linear model (CSGLM), adjusting for all covariates in association with grip strength. The collected raw data of the KNHANES are not a complete enumeration survey but rather a sample survey with a complex sampling design, which means that sample data do not have an equal probability of being selected. In addition, sample data have the unequal response and dropout rates. Therefore, it is recommended to use sampling weights when analyzing the data and estimating results for the target population. The sampling weight was calculated from non-response adjusted weights using estimated response probability, and calibration weights, accounting for the clustering and stratification of the sample survey data. Interpretation of ICC values was based on guidelines offered by Portney and Watkins [20], with values above 0.75 being classified as good reliability, and those below 0.75 being classified as moderate to poor reliability. Coefficients of variation (CVs, %) were also calculated to assess the absolute reliability across grip measurements. *p* values < 0.05 were considered statistically significant. All statistics were calculated using Statistical Package for the Social Sciences version 22 (IBM/SPSS Inc., Armonk, NY, USA).

## 3. Results

### 3.1. Descriptive Statistics

The total number of participants aged over 10 years was 6577, including 3062 males and 3515 females. The mean age of 6577 participants was 43.63 ± 0.34 years (95% CI = 42.96–44.30) in males and 43.67 ± 0.34 years (95% CI = 43.01–44.33) in females. The participants showed a higher proportion of right-hand preference (90% in both male and female). In occupation, the participants of the occupation group with high physical demand were 40.1 % in male participants and 21.0 % in female participants. The proportion of job seeker in total participant was 6.6 % in 10–19 age level, 30.5% in 20–29, 19.0% in 30–39, 19.5% in 40–49, 22.2% in 50–59(8.2%: retired), 20.2% in 60–69(17.5%: retired), 13.5% in ≥70 (16.3%: retired). Even though older participants responded that they have no problem in daily or social activities in self-reported questionnaire and EQ-5D, underlying medical conditions was detected when measured by trained medical staff and the proportion of underlying medical conditions was rising in middle age level. The general characteristics of the participants are presented in Table 1 and Table 2 in more detail.

### 3.2. Covariates Effects

#### 3.2.1. Unmodifiable Covariates (Gender, Age and Hand Preference) Effect

The mean of grip strength in total 6577 participants was 33.54 ± 0.16 kg (95% CI: 33.22–33.86) in right hand and 31.92 ± 0.16 kg (95% CI: 31.61–32.22) in left hand. The mean of grip strength in male participants was 41.04 ± 0.18 kg (95% CI: 40.68–1.40) in right hand and 39.27 ± 0.18 kg (95% CI: 38.92–39.62) in left hand, which was significantly more great than females (24.77 ± 0.12 kg (95% CI: 24.54–25.00, effect size: 1.35; *p* = 0.001) in right; 23.34 ± 0.16 kg (95% CI: 23.11–23.56, effect size: 1.34; *p* = 0.001) in left.

We analyzed each unmodifiable factor in association with grip strength. Male participants of 30–39 age level had 44.49 ± 0.37 kg (95% CI: 43.76–45.23) in the right hand and 42.39 ± 0.36 kg (95% CI: 41.69–43.09) in the left hand (30–39 age level). In the female of 30–39 age level, the grip strength was 26.06 ± 0.21 kg (95% CI: 25.67–26.48) in the right hand and 24.58 ± 0.20 kg (95% CI: 24.18–24.98) in the left hand. The grip strength was greatest in 30–39 age level of male group and 40–49 age level of female group, when stratified by age level (decade), but was lowest in over 70 age level in both genders (overall effect size: 1.49; *p* = 0.001 in male, overall effect size: 0.76; *p* = 0.001 in female). The grip strength of each male group was significantly stronger than females for both hands in all age levels (effect size: 1.72–2.77; *p* = 0.001).

In both genders, the effect of hand preference on grip strength showed that right grip strength in right hand preference is stronger than right grip strength in left hand preference, which showed that left grip of left hand preference also had stronger than left grip strength in right hand preference, however, which was not statistically insignificant (*p* = 0.08).

#### 3.2.2. Modifiable Covariates (Anthropometric Factors, Occupation and Underlying Medical Condition) Effect

An increment of 1 kg in weight was related to a 0.5 kg increase in grip strength of both hands (R: 0.62; *p* = 0.001), and an increment of 1 cm in height was related to a 0.82 kg increase in the right-hand grip and 0.80 kg in the left-hand grip (R: 0.72; *p* = 0.001). An increment of 1 cm in waist circumference was related to a 0.40 kg increase in both grip strengths (R: 0.39; *p* = 0.001).

Occupation also was related to grip strength. The mean strengths of both hands in subjects with more physically demanding occupations were 7.5 kg greater than those of subjects with occupations with low physical demands and 1.9 kg greater than those of subjects with medium physical demands (R: 0.32; *p* = 0.001).

The grip strengths of participants with HBP were 0.85 kg greater in the left hand than in participants without HBP (R < 0.03; *p* = 0.02). The strengths of participants without OA were 7.0 kg greater in both hands than in participants without OA (R: 0.08; *p* = 0.001). Strengths did not significantly differ according to DM status (R < 0.03; *p* = 0.20).

### 3.3. Multivariate Analysis

Because sample data have the unequal probability of being selected and response and dropout rate, the normative value of grip strength should be analyzed and estimated while adjusting for all covariates in association with grip strength.

After adjusting for all included covariates, the greatest grip strength was 42.07 ± 0.82 kg (95% CI: 40.45–43.69) in the right hand and 41.06 ± 0.75 kg (95% CI: 39.59–42.53) in the left hand of males aged 30–39 years old. In females, those aged 40–49 years old had the greatest grip strengths, at 25.98 ± 0.44 kg (95% CI: 25.12–26.84) in the right hand and 25.33 ± 0.44 kg (95% CI: 24.46–26.2) in the left hand (Table 3 and Table 4). Grip strength showed a curvilinear trend with those aged 70 years old and over. Grip strength was also low in those aged 10–19 years old. An increment of 1 kg in weight was related to an increase of 0.3 kg strength in both hands, and an increment of 1 cm in height was related to an increase of 0.1 kg strength in both hands (*p* = 0.001). Participants with occupations with high physical demands had a grip strength that was 1.3 kg and 2.1 kg greater in both hands than the medium and low physical demand groups (*p* = 0.001), respectively.

Interestingly, the results differed before and after adjusting for all covariates. The right-hand grip strength was 1.5 kg greater in right-handed participants than in left-handed participants, and the left-hand grip strength was 1.6 kg greater in left-handed participants than in right-handed participants (*p* = 0.001). An increment of 1 cm waist circumference was inversely associated with a 0.3 kg decrease in grip strength (*p* = 0.001). The strengths of participants without DM were 0.7 kg and 1.2 kg greater in the right (*p* = 0.025) and left (*p* = 0.001) hands, respectively, than in participants with DM. The strengths of those with HBP were 0.1 kg less than those without HBP (*p* = 0.58), and the strengths of those with OA were 0.3 kg less than those without OA (*p* = 0.50); however, the differences in these two measures were not statistically significant.

To predict grip strength, we made the regression equation that fit data, using complex-samples general linear model (CSGLM) analysis. The formula for estimating grip strength was demonstrated in Table 5.

## 4. Discussion

Normative data on grip strength have been published by many authors, but such studies have been limited to a narrow age range of individuals [21,22,23], or included participants who were not perfectly representative of the population [24]. Many potential confounding factors associated with grip strength have also been reported but have not been adjusted for, limiting the ability of the results to represent a reference value of grip strength.

The definition for “healthy participants” included in the previous studies was also somewhat different from previous studies, which have enrolled subjects using many different criteria, such as literally “healthy”, “having neither cognitive impairment nor physical impairment of the upper extremity”, and “no neurological deficit”. In the present study, healthy participants were defined using EuroQol 5-Dimension Questionnaire (EQ-5D), not merely the absence of disease or infirmity.

After adjusting for all covariates, grip strength in both hands was highest in those aged 30–49 years old in both sexes, revealing a curvilinear relationship between grip strength and age. Males had stronger grips than females. Our results are consistent with the international norms that grip strength declines with age and is stronger in males. However, the values in our study were lower than internationally published norms [22,23,25,26,27,28,29], which may be due to the difference in participants sampling, grip measurement and included covariates.

Anthropometric factors are associated with grip strength. Changes in weight, height, and waist circumference were associated with changes in grip strength. However, after adjusting for all covariates using CSGLM, a 1 cm increase in waist circumference was associated with a 0.3 kg decrease in grip strength. Similar results have been reported in previous studies [30].

We also found that DM affected grip strength, consistent with a previous study [31]. The grip strength of participants without DM was greater 0.7 kg in right hand and 1.2 kg in the left hand than participants with DM. Indeed, neuronal changes associated with diabetes or glucose intolerance may induce a change in grip strength [31,32].

In our study, participants with occupations with high physical demands had greater grip strength than other groups, which suggests that lifestyle may be associated with grip strength. A previous study considered this issue controversial [33].

We also found that the right grip strengths of right-handed participants were 1.5 kg greater than those of left-handed participants, whereas the strengths of the left hands of left-handed individuals were 1.6 kg greater than those of right-handed individuals. These results are not in line with the “10% rule” that the dominant hand possesses a 10% greater grip strength than the nondominant hand [34].

Previous studies have suggested other potential factors associated with grip strength. Handgrip strength or performance is a key function used in daily activities, therefore, it may be used as an indicator of whole-body muscle strength. Reduced handgrip strength or performance may represent a decrement in human activity function, which may result from age- or disease-related processes. Previous studies have reported that reduced grip strength may be associated with the cognitive decline with aging, even in cognitively healthy elderly [9,35]. Poor nutritional status is also associated with poor handgrip strength [36,37]. Therefore, to make a more accurate estimate of grip strength that is perfectly representative of the population, there are still potential factors associated with grip strength to consider.

To estimate grip strength with different anthropometric and background characteristics, we tried to include factors that may be associated with grip strength. However, our study had some limitations. First, it is not possible to infer a causal relationship between these factors and observed trends due to the cross-sectional design. The possibility of sample selection bias and endogeneity bias is present. There may have also been inter-rater variation in grip strength measurements, although the data were analyzed using sample weights. The high number of subjects (*n* = 6577) may help overcome bias. However, our results included data from just one nation in Asia. Thus, our results have an endogenous bias regarding ethnic homogeneity. The limitations for the survey itself as mentioned may be a disturbing factor for our study. For estimating grip strength, we suggest the formula for grip strength with a functional constant for future users. However, more comprehensive, international, and multi-ethnic studies with valid testers are needed to address whether the estimated formula can predict grip strength in different ethnic groups.

## 5. Conclusions

In our results, occupation, waist circumference, and DM may affect grip strength in our nationwide population-based investigation. To determine normative data for grip strength, factors such as occupation, underlying medical conditions, anthropometric characteristics, and unmodifiable factors such as age and sex should be considered. To predict grip strength, we generated a regression equation that fit the data. Our formula may be useful for predicting or identifying functional decrements in grip strength in patients for whom functional impairment is expected.

## Figures and Tables

**Table 1 ijerph-16-02235-t001:** The General Characteristics of Participants (*n* = 6577).

Age Level (years)	Male (*n* = 3062)				
	Right Preference (*n* (%))	High ^†^ (*n* (%))	HBP (*n* (%))	DM (*n* (%))	Arthritis (*n* (%))
**10–19 (*n* = 58)**	53 (90.1%)	12 (26.01%)	2 (2.7%)	0 (0.0%)	0 (0.0%)
**20–29 (*n* = 382)**	346 (90.5%)	88 (21.97%)	20 (4.4%)	2 (0.4%)	0 (0.0%)
**30–39 (*n* = 517)**	463 (90.3%)	186 (36.67%)	81 (14.9%)	13 (2.5%)	0 (0.0%)
**40–49 (*n* = 585)**	512 (89.9%)	243 (45.12%)	156 (26.6%)	43 (8.5%)	2 (0.4%)
**50–59 (*n* = 609)**	533 (88.5%)	350 (56.44%)	211 (34.6%)	73 (11.7%)	10 (1.1%)
**60–69 (*n* = 530)**	483 (91.2%)	266 (52.00%)	264 (48.2%)	130 (24.9%)	12 (2.4%)
**≥70 (*n* = 381)**	318 (82.9%)	107 (26.07%)	212 (54.5%)	71 (20.9%)	20 (6.0%)
**Age Level** **(years)**	**Female (*n* = 3515)**				
	**Right** **Preference (*n* (%))**	**High** **^†^** **(*n* (%))**	**HBP** **(*n* (%))**	**DM** **(*n* (%))**	**Arthritis** **(*n* (%))**
**10–19 (*n* = 52)**	46 (85.5%)	6 (13.26%)	0 (0.0%)	0 (0.0%)	0 (0.0%)
**20–29 (*n* = 459)**	421 (92.7%)	43 (9.68%)	3 (0.4%)	0 (0.0%)	0 (0.0%)
**30–39 (*n* = 679)**	599 (89.8%)	69 (10.52%)	11 (1.3%)	18 (2.6%)	3 (0.7%)
**40–49 (*n* = 747)**	659 (90.1%)	167 (22.99%)	78 (11.2%)	28 (4.3%)	4 (0.8%)
**50–59 (*n* = 744)**	632 (87.0%)	267 (36.43%)	210 (28.7%)	55 (8.1%)	30 (5.1%)
**60–69 (*n* = 511)**	452 (89.7%)	166 (32.46%)	250 (50.1%)	63 (14.9%)	41 (8.8%)
**≥70 (*n* = 323)**	273 (87.4%)	68 (19.67%)	224 (68.5%)	63 (22.9%)	41 (14.6%)

Values (*n* (%)) are the number and percentage of participants. ^†^: occupation group with high physical demand. HBP: hypertension; DM: diabetes mellitus.

**Table 2 ijerph-16-02235-t002:** The Anthropometric Characteristics of Participants (*n* = 6577).

Age Level (years)	Male (*n* = 3062)		
	Height (cm)	Weight (kg)	Waist Circumference (cm)
**10–19**	173.15 ± 0.6 (171.96–174.33)	67 ± 1.41 (64.23–69.77)	77.93 ± 1.17 (75.62–80.24)
**20–29**	174.04 ± 0.33 (173.39–174.69)	71.77 ± 0.66 (70.47–73.08)	81.74 ± 0.54 (80.68–82.81)
**30–39**	174.83 ± 0.27 (174.3–175.36)	76.00 ± 0.53 (74.95–77.05)	85.95 ± 0.38 (85.2–86.71)
**40–49**	172.35 ± 0.25 (171.85–172.85)	74.18 ± 0.44 (73.31–75.06)	86.42 ± 0.36 (85.72–87.13)
**50–59**	169.51 ± 0.26 (168.99–170.03)	70.18 ± 0.4 (69.4–70.96)	86.09 ± 0.34 (85.43–86.76)
**60–69**	166.8 ± 0.27 (166.28–167.32)	67.45 ± 0.42 (66.63–68.27)	86.55 ± 0.37 (85.84–87.27)
**≥70**	165.01 ± 0.36 (164.31–165.71)	63.16 ± 0.5 (62.17–64.15)	85.31 ± 0.53 (84.28–86.34)
**Age level** **(years)**	**Female (*n* = 3515)**		
	**Height** **(cm)**	**Weight** **(kg)**	**Waist Circumference** **(cm)**
**10–19**	160.96 ± 0.68 (159.62–162.3)	56.12 ± 1.86 (52.47–59.77)	71.82 ± 1.56 (68.75–74.88)
**20–29**	161.64 ± 0.32 (161.01–162.28)	56.29 ± 0.58 (55.14–57.44)	72.11 ± 0.51 (71.1–73.12)
**30–39**	161.07 ± 0.23 (160.63–161.52)	58.65 ± 0.42 (57.83–59.48)	76.25 ± 0.41 (75.44–77.06)
**40–49**	158.7 ± 0.21 (158.29–159.1)	58.2 ± 0.33 (57.56–58.85)	77.09 ± 0.34 (76.41–77.76)
**50–59**	156.38 ± 0.21 (155.96–156.79)	57.98 ± 0.31 (57.36–58.6)	79.19 ± 0.37 (78.46–79.91)
**60–69**	154.56 ± 0.28 (154.01–155.11)	57.72 ± 0.38 (56.98–58.46)	81.54 ± 0.47 (80.61–82.47)
**≥70**	151.19 ± 0.44 (150.31–152.06)	55.21 ± 0.63 (53.97–56.44)	83.5 ± 0.59 (82.34–84.65)

Values are numbers and depicted as the mean ± standard error (95% confidence interval).

**Table 3 ijerph-16-02235-t003:** The Normative Value of Grip Strength Adjusted for All Covariates in Male.

Age Level (years)	Right Grip (kg)	Left Grip (kg)
**10–19**	35.76 ± 1.21 (33.37–38.14)	34.5 ± 1.08 (32.37–36.63)
**20–29**	39.25 ± 0.8 (37.67–40.82)	38.24 ± 0.72 (36.82–39.66)
**30–39**	42.07 ± 0.82 (40.45–43.69)	41.06 ± 0.75 (39.59–42.53)
**40–49**	41.67 ± 0.75 (40.19–43.15)	40.96 ± 0.67 (39.65–42.27)
**50–59**	40.24 ± 0.73 (38.8–41.68)	39.86 ± 0.64 (38.59–41.13)
**60–69**	38.8 ± 0.7 (37.42–40.17)	38.31 ± 0.65 (37.03–39.6)
**≥70**	35.14 ± 0.76 (33.64–36.65)	35.04 ± 0.65 (33.76–36.31)

Values are numbers and depicted as the mean ± standard error (95% confidence interval).

**Table 4 ijerph-16-02235-t004:** The Normative Value of Grip Strength Adjusted for All Covariates in Female.

Age Level (years)	Right Grip (kg)	Left Grip (kg)
**10–19**	24.61 ± 0.85 (22.95–26.28)	23.73 ± 1.01 (21.75–25.72)
**20–29**	23.53 ± 0.48 (22.59–24.47)	22.71 ± 0.5 (21.74–23.69)
**30–39**	25.65 ± 0.42 (24.82–26.48)	24.95 ± 0.45 (24.07–25.83)
**40–49**	25.98 ± 0.44 (25.12–26.84)	25.33 ± 0.44 (24.46–26.2)
**50–59**	25.72 ± 0.42 (24.89–26.55)	24.72 ± 0.42 (23.89–25.55)
**60–69**	24.56 ± 0.42 (23.73–25.39)	23.78 ± 0.43 (22.94–24.62)
**≥70**	21.98 ± 0.59 (20.81–23.15)	22.05 ± 0.54 (20.99–23.11)

Values are numbers and depicted as the mean ± standard error (95% confidence interval).

**Table 5 ijerph-16-02235-t005:** The Suggested Formula for Estimating Grip Strength.

Right Hand Grip	2.71 + 11.94 × (gender) + (age) + (hand preference) + (occupation) + 0.25 × HBP + 0.67 × OA − 0.45 × DM+0.11 × height + 0.32 × weight − 0.23 × (waist circumference) (R = 0.85)
**Functional constant**	(gender) male = 1, female = 0 (age) 10–19 age level = 1.926, 20–29 age level = 3.382, 30–39 age level = 6.004, 40–49 age level = 5.858, 50–59 age level = 4.886, 60–69 age level = 3.484, over 70 age = 0 (hand preference) right hand preference = −0.05, left hand preference = −1.47, ambidexterity = 0 (occupation): heavy = 1.3, medium = −0.001, light = 0 (HBP) HBP = 1, normal = 0; (OA) OA = 1, no OA = 0; (DM) DM = 1, no DM = 0
**Left Hand Grip**	**5.43 + 11.79 × (gender) + (age) + (hand preference) + (occupation) + 0.42 × HBP + 0.474 × OA − 0.937 × DM + 0.091 × height+0.336 × weight − 0.25 × (waist circumference) (R = 0.85)**
**Functional constant**	(gender) male = 1, female = 0 (age) 10–19 age level = 0.916, 20–29 age level = 2.574, 30–39 age level = 5.25, 40–49 age level = 5.281, 50–59 age level = 4.308, 60–69 age level = 2.918, over 70 age = 0 (hand preference) right hand preference = −1.114, left hand preference = 0.589, ambidexterity = 0 (occupation): heavy = 1.25, medium = −0.047, light = 0 (HBP) HBP = 1, normal = 0; (OA) OA = 1, no OA = 0; (DM) DM = 1, no DM = 0

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
