# Peer review of "Normative Data on Grip Strength in a Population-Based Study with Adjusting Confounding Factors: Sixth Korea National Health and Nutrition Examination Survey (2014–2015)"

_ijerph, 2019, doi:10.3390/ijerph16122235_

Round 1

Reviewer 1 Report

ijerph-512144 presented interesting results from the KNHANES. While some parts of this manuscript were interesting, other areas could be improved. I hope the authors consider my suggestions for enhancing their manuscript.

MAJOR COMMENTS

Lines 117-125: I am not sure if this information is relevant for the measures section. As currently presented, the information is more so relevant for the statistical analyses and results sections. Consider revision here to address this.

Lines 147-153: Operational definitions here need a supporting citation.

Results: Many of the p-values listed were “.00”. While I understand that the authors were probably rounding these values, it may be difficult for the reader to interpret a p-value of zero. Can the authors revise such p-values?

Section 3.2.2: These analyses were not clearly presented in the statistical analyses section. Please provide more information here. Similarly, the authors should consider presenting the r-value instead of r-squared. While r-squared presents the variance explained, it does not reveal direction (r-values do). The reader can calculate r-squared from r-values.

Results: It is challenging to include diseases such as OA and DM in these formulas (and results) given the study design. For example, in a cross-sectional investigation, factors related to neuropathy make it challenging to determine if weakness contributes to DM of if DM contributes to weakness. This is acknowledged as a limitation but is certainly impactful for the results. Language needs to be tempered in the discussion where appropriate.

MINOR COMMENTS

Lines 34-35: Should be, “Measures of grip strength are an important for geriatric and occupational assessments” or similar. Be sure to correct any grammatical errors for English language throughout the manuscript.

Lines 43-44: Can you revise to make the purpose statement clearer? It is not really a purpose statement in its current form.

Line 50: “It was” please be more specific about this. Again, please correct any grammatical errors throughout the manuscript.

Lines 74-75: This statement needs a citation for support.

Line 100: No need to mention the Jamar dynamometer if not used.

Line 127: Should be “statistical analyses.”

Line 174-175: This information was already presented.

Tables: Make sure all tables stand alone.

Make changes to the abstract that align with the text.

Author Response

Manuscript ID: ijerph-512144

Title: Normative data for grip strength in a population-based study with adjustment for confounding factors: The Sixth Korea National Health and Nutrition Examination Survey (2014-2015)

Dear reviewer

The authors are grateful for the insightful and valuable comments the reviewers have made to our paper “Normative data for grip strength in a population-based study with adjustment for confounding factors: The Sixth Korea National Health and Nutrition Examination Survey (2014-2015)”. We feel confident that your comments would contribute in raising the overall academic standards and clinical significance of this work. Each comment made from the reviewers was thoroughly discussed with the other co-authors. We went through each single comment with great scrutiny and made sure that we made the appropriate corrections to the manuscript as suggested by the reviewers.

Reviewer #1:

MAJOR COMMENTS

Lines 117-125: I am not sure if this information is relevant for the measures section. As currently presented, the information is more so relevant for the statistical analyses and results sections. Consider revision here to address this.

Comment:  Lines 117-125 were moved to ‘Data analysis section’ in line 150-152 as recommended.

Lines 147-153: Operational definitions here need a supporting citation.

Comment: We added the references as recommended.

Results: Many of the p-values listed were “.00”. While I understand that the authors were probably rounding these values, it may be difficult for the reader to interpret a p-value of zero. Can the authors revise such p-values?

Comment: P = .00 was revised as P = .001 as recommended.

Section 3.2.2: These analyses were not clearly presented in the statistical analyses section. Please provide more information here. Similarly, the authors should consider presenting the r-value instead of r-squared. While r-squared presents the variance explained, it does not reveal direction (r-values do). The reader can calculate r-squared from r-values.

Comment: We changed r-squared to r-value as recommended.

Results: It is challenging to include diseases such as OA and DM in these formulas (and results) given the study design. For example, in a cross-sectional investigation, factors related to neuropathy make it challenging to determine if weakness contributes to DM of if DM contributes to weakness. This is acknowledged as a limitation but is certainly impactful for the results. Language needs to be tempered in the discussion where appropriate.

Comment: We agree with the opinion of reviewer. The problem to infer causality the direction of the effect due to the cross-sectional design was described as recommended in line 285-292.

MINOR COMMENTS

Lines 34-35: Should be, “Measures of grip strength are an important for geriatric and occupational assessments” or similar. Be sure to correct any grammatical errors for English language throughout the manuscript.

Comment: We changed the sentences as recommended in line 33-34.

The English in this document has been checked by at least two professional editors, both native speakers of English. For a certificate, please see:

http://www.textcheck.com/certificate/T0hAEY

Lines 43-44: Can you revise to make the purpose statement clearer? It is not really a purpose statement in its current form.

Comment: We changed the sentences as recommended in line 40-42.

Line 50: “It was” please be more specific about this. Again, please correct any grammatical errors throughout the manuscript.

Comment: Two reviewers’ opinion for that sentence were different. Thus, we decided “It was” with references (our previous publications). We appreciate your comments.

Ref>

1. Kwon, K.M.; Lee, J.S.; Jeon, N.E.; Kim, Y.H. Factors associated with health-related quality of life in Koreans aged over 50 Years: the fourth and fifth Korea National Health and Nutrition Examination Survey. Health Qual Life Outcomes 2017, 15, 243, doi:10.1186/s12955-017-0816-4.

2. Kim, Y.H.; Lee, J.S.; Park, J.H. Association between bone mineral density and knee osteoarthritis in Koreans: The Fourth and Fifth Korea National Health and Nutrition Examination Surveys. Osteoarthritis Cartilage 2018, 26, 1511-1517, doi:10.1016/j.joca.2018.07.008.

Lines 74-75: This statement needs a citation for support.

Comment: We added the reference as recommended.

Line 100: No need to mention the Jamar dynamometer if not used.

Comment: We removed the sentences from recommended in line 92-95.

Line 127: Should be “statistical analyses.”

Comment: We changed the sentences as recommended in line 111-112.

Line 174-175: This information was already presented.

Comment: We rechecked and modified the results and discussion

Tables: Make sure all tables stand alone.

Comment: We divided the Table 3 to table 3 and 4, as recommended.

Make changes to the abstract that align with the text.

Comment: We modified the abstract that align with the text as recommended.

We would like to thank the reviewers for their insightful comments to help overcome the many shortcomings of our study. We feel confident that these comments would help to upgrade the overall quality of our work. The authors would always be ready to answer to any further comments from the Editors and the reviewers.

Best Regards;

Corresponding author: Jung Soo Lee, M.D., Ph.D.

Reviewer 2 Report

In the current studies the authors evaluated grip strength in a large population of adults.  The sample size was a strength of the study.  However, there were substantial grammatical, spelling, and, sentence construction issues throughout the entire manuscript.  These require correction before this paper can be appropriately reviewed.  The novelty and clinical relevance of this paper are also low.

Author Response

Manuscript ID: ijerph-512144

Title: Normative data for grip strength in a population-based study with adjustment for confounding factors: The Sixth Korea National Health and Nutrition Examination Survey (2014-2015)

Dear reviewer

The authors are grateful for the insightful and valuable comments the reviewers have made to our paper “Normative data for grip strength in a population-based study with adjustment for confounding factors: The Sixth Korea National Health and Nutrition Examination Survey (2014-2015)”. We feel confident that your comments would contribute in raising the overall academic standards and clinical significance of this work. Each comment made from the reviewers was thoroughly discussed with the other co-authors. We went through each single comment with great scrutiny and made sure that we made the appropriate corrections to the manuscript as suggested by the reviewers.

Reviewer #2

In the current studies the authors evaluated grip strength in a large population of adults.  The sample size was a strength of the study.  However, there were substantial grammatical, spelling, and, sentence construction issues throughout the entire manuscript.  These require correction before this paper can be appropriately reviewed.  The novelty and clinical relevance of this paper are also low.

Comments:

1. I modified Introduction, Methods and discussion as recommended.

2. We checked and corrected the grammatical, spelling, and, sentence construction by two native speakers of English before final revision.

The English in this document has been checked by at least two professional editors, both native speakers of English. For a certificate, please see:

http://www.textcheck.com/certificate/T0hAEY

We would like to thank the reviewers for their insightful comments to help overcome the many shortcomings of our study. We feel confident that these comments would help to upgrade the overall quality of our work. The authors would always be ready to answer to any further comments from the Editors and the reviewers.

Best Regards;

Corresponding author: Jung Soo Lee, M.D., Ph.D.

Reviewer 3 Report

The authors present an interesting analysis of grip strength together with a panel of other indicators among 6577 participants to the KNHANES survey. Grip strength is considered a good  an indicator of overall muscular strength, and the interesting point here is that this parameters is matched with health, nutritional status, and physical activity parameters. The main aim here, as in other studies on this topic, is to provide normative values and predictive formulas based on a model taking into account the relative weight of various confounding factors. Thus representative data and reference equations for the Korean population (over 10 years of age) are provided, and these equations can be used to determine the reference value for a person, given other information: age, sex, height, weight and daily activities. 

However the paper needs to be improved in several aspects and at this stage is not recommended for publication. 

In general:

-   the authors initially claim that they considered only healthy participants to the survey, but later in the manuscript they review this statement in the light of the information gathered and collected. Therefore it would be advisable to more and more accurate information on the characteristics of the participants since the beginning.

form the methods (Data analysis) it’s not really clear how the model equations were developed. The authors should be a little more detailed and specific about this.

the authors see a limit of this analysis in the fact that it is limited to participants of a single nation. However similar studies have also been conducted in other countries, albeit with some different accompanying data. It would be useful if the authors put their results in relation to other studies, in addition to recommending the need for international studies. Another limitation is that there is no reported accurate measure of physical activity of participants, but of course this is understandable given the number of participants.

some sentences must be revised to give more clarity and fluency.

Abstract

Line 17 Can usual be changed to daily?

Line 23-24  Could the authors change “were” with “are” in the sentence: “The formulas for estimating grip strength of each hand were presented as main results.”

Line 24-27 Can this sentence be made clearer to the reader?

Introduction

Line 31 Can has been be changed to is? 

Line 35-36  The grip strength is an expression/assesment of muscle fiction. Can this sentence be made clearer?

Line 43-44  Could this sentence be made clearer?

Methods

Line 50-52 I suggest to change this sentence in:”It was a transversal national health examination and survey to have a representative description of the state of health, nutritional status and physical activity of the Korean population in general.”  

Results

Line 226-228 Can this statement be made clearer?

Lines 222-233 Can this statement be made clearer? Here it’s not clear how data are compared: on the basis of means; adjusted values or correlations?

Discussion

Line 296-298 Can the authors explain how the decline of grip strength could contribute to the decline of cognitive capacities? There would appear to be a causal link between strength reduction and loss of cognitive abilities…..or perhaps these are two parallel phenomena for which it is difficult to disentangle whether the origin is central or peripheral?… or maybe it is possible and known. 

Line 304-306 It’ true that this is a cross-sectional design and it’s not thought to produce explanatory results. However it could be interesting to divide the participants in three groups according to handgrip performance and compare the highest and the lowest. This could allow some considerations.

Line 309-310  Can the authors change “Thus, our results had an endogeneity bias for racial homogeneity.” to “Thus, our results have an endogenous bias concerning ethnic homogeneity.”

Line 310-311 Can this sentence made clearer?

Line 314 Can “multi-nation, multi-racial population-based …“ be changed to “more comprehensive,international and multi ethnical studies…”

Conclusions

Line 325-326 It’s not clear what the authors mean for “neurological management strategies”. Make this sentence clearer.

Author Response

Manuscript ID: ijerph-512144

Title: Normative data for grip strength in a population-based study with adjustment for confounding factors: The Sixth Korea National Health and Nutrition Examination Survey (2014-2015)

Dear reviewer

The authors are grateful for the insightful and valuable comments the reviewers have made to our paper “Normative data for grip strength in a population-based study with adjustment for confounding factors: The Sixth Korea National Health and Nutrition Examination Survey (2014-2015)”. We feel confident that your comments would contribute in raising the overall academic standards and clinical significance of this work. Each comment made from the reviewers was thoroughly discussed with the other co-authors. We went through each single comment with great scrutiny and made sure that we made the appropriate corrections to the manuscript as suggested by the reviewers.

Reviewer #3

the authors present an interesting analysis of grip strength together with a panel of other indicators among 6577 participants to the KNHANES survey. Grip strength is considered a good an indicator of overall muscular strength, and the interesting point here is that these parameters is matched with health, nutritional status, and physical activity parameters. The main aim here, as in other studies on this topic, is to provide normative values and predictive formulas based on a model taking into account the relative weight of various confounding factors. Thus representative data and reference equations for the Korean population (over 10 years of age) are provided, and these equations can be used to determine the reference value for a person, given other information: age, sex, height, weight and daily activities.

However, the paper needs to be improved in several aspects and at this stage is not recommended for publication.

In general:

-   the authors initially claim that they considered only healthy participants to the survey, but later in the manuscript they review this statement in the light of the information gathered and collected. Therefore it would be advisable to more and more accurate information on the characteristics of the participants since the beginning.

Comment: We added and modified the information of the participants as recommended in line 63-78 and line 251-252.

- form the methods (Data analysis) it’s not really clear how the model equations were developed. The authors should be a little more detailed and specific about this.

Comment: We described more detailed as recommended in line 237-238.

- the authors see a limit of this analysis in the fact that it is limited to participants of a single nation. However similar studies have also been conducted in other countries, albeit with some different accompanying data. It would be useful if the authors put their results in relation to other studies, in addition to recommending the need for international studies. Another limitation is that there is no reported accurate measure of physical activity of participants, but of course this is understandable given the number of participants.

Comment: We modified the limitation of ‘discussion’ section in line 284-292, as recommended.

- some sentences must be revised to give more clarity and fluency.

Comment: We checked and corrected the grammatical, spelling, and, sentence construction by two native speakers of English before final revision.

The English in this document has been checked by at least two professional editors, both native speakers of English. For a certificate, please see:

http://www.textcheck.com/certificate/T0hAEY

Abstract

Line 17 Can usual be changed to daily?

Comment: We changed the sentences as recommended in line 17.

Line 23-24 Could the authors change “were” with “are” in the sentence: “The formulas for estimating grip strength of each hand were presented as main results.”

Comment: We changed the sentences as recommended in line 23.

Line 24-27 Can this sentence be made clearer to the reader?

Comment: We changed the sentences as recommended in line 24-26.

Introduction

Line 31 Can has been be changed to is?

Comment: We changed the sentences as recommended in line 30.

Line 35-36  The grip strength is an expression/assessment of muscle fiction. Can this sentence be made clearer?

Comment: We changed the sentences as recommended in line 33-34.

Line 43-44  Could this sentence be made clearer?

Comment: We changed the sentences as recommended in line 40-42.

Methods

Line 50-52 I suggest to change this sentence in:”It was a transversal national health examination and survey to have a representative description of the state of health, nutritional status and physical activity of the Korean population in general.” 

Comment: We changed the sentences as recommended in line 46-48.

Results

Line 226-228 Can this statement be made clearer?

Comment: We changed the sentences as recommended in line 206-209.

Lines 222-233 Can this statement be made clearer? Here it’s not clear how data are compared: on the basis of means; adjusted values or correlations?

Comment: We changed the sentences as recommended in line 202-205.

Discussion

Line 296-298 Can the authors explain how the decline of grip strength could contribute to the decline of cognitive capacities? There would appear to be a causal link between strength reduction and loss of cognitive abilities…..or perhaps these are two parallel phenomena for which it is difficult to disentangle whether the origin is central or peripheral?… or maybe it is possible and known.

Comment: We added the explanations with references as recommended in line 275-283.

Line 304-306 It’ true that this is a cross-sectional design and it’s not thought to produce explanatory results. However, it could be interesting to divide the participants in three groups according to handgrip performance and compare the highest and the lowest. This could allow some considerations.

Comment: We added two multivariate logistic regression models considering the highest grip strength group as recommended. (For review only)

Multivariate logistic regression was used to estimate the odds ratio (OR) and 95% confidence intervals (CIs) according to quartiles of each grip strength. Model 1 was adjusted for gender, age (years), hand preference, height, weight, waist circumference. Model 2 was further adjusted for gender, age (years), hand preference, height, weight, waist circumference, occupation, hypertension, osteoarthritis, diabetes mellitus. A level of 5% was used as the level of significance.

Table1. Multivariate logistic regression model 1 considering the highest grip strength group

OR(95%CI)

in   right hand

P-value

OR(95%CI)

in   left hand

P-value

Gender

(reference: female)

Male

3187.55

(823.32 to 12340.82)

0.0001

14.40

(8.71 to 23.79)

0.0001

Age

(reference: over 70 years)

10 to 20 years

1.03

(0.11 to 9.58)

0.97

1.86

(0.91 to 3.79)

0.08

20 to 30 years

8.35

(1.75 to 39.73)

0.008

1.22

(0.80 to 1.86)

0.34

30 to 40 years

29.17

(2.92 to 290.71)

0.004

3.61

(2.36 to 5.51)

0.0001

40 to 50 years

80.03

(19.38 to 330.52)

0.000

4.16

(2.84 to 6.10)

0.0001

50 to 60 years

81.05

(13.61 to 482.40)

0.000

3.07

(2.14 to 4.39)

0.0001

60 to 70 years

33.01

(9.19 to 118.49)

0.000

2.13

(1.49 to 3.05)

0.0001

Hand   preference

(reference:   bimanual)

right

3.48

(1.08 to 11.25)

0.03

0.81

(0.54 to 1.22)

0.32

left

1.96

(0.29 to 12.87)

0.48

1.04

(0.57 to 1.89)

0.89

Height

1.28

(1.18 to 1.40)

0.0001

1.04

(1.02 to 1.06)

0.0001

Weight

1.14

(1.02 to 1.27)

0.01

1.09

(1.06 to 1.12)

0.0001

Waist   circumference

0.92

(0.81 to 1.04)

0.20

0.94

(0.92 to 0.97)

0.0001

Table2. Multivariate logistic regression model 2 considering the highest grip strength group

OR(95%CI)

in   right hand

P-value

OR(95%CI)   in left hand

P-value

Gender

(reference: female)

male

11838.01

(2690.99 to 52076.88)

0.000001

17.87

(9.77 to 32.68)

0.00000001

Age

(reference: over 70 years)

10 to 20 years

0.72

(0.11 to 4.71)

0.73

2.10

(0.87 to 5.05)

0.09

20 to 30 years

6.49

(1.20 to 35.07)

0.02

1.25

(0.72 to 2.19)

0.42

30 to 40 years

70.83

(6.076 to 825.66)

0.0007

3.88

(2.23 to 6.75)

0.000001

40 to 50 years

178.47

(31.78 to 1002.10)

0.0000

4.27

(2.56 to 7.14)

0.0001

50 to 60 years

135.06

(7.94 to 2295.10)

0.0007

3.39

(2.08 to 5.51)

0.0001

60 to 70 years

33.30

(6.34 to 174.84)

0.00001

1.95

(1.22 to 3.10)

0.004

Hand   preference

(reference:   bimanual)

right

7.60

(2.19 to 26.38)

0.001

0.81

(0.53 to 1.24)

0.34

left

3.54

(0.28 to 44.13)

0.32

1.03

(0.54 to 1.99)

0.91

Occupation

(reference:   light)

medium

2.50

(0.43 to 14.44)

0.30

0.92

(0.72 to 1.18)

0.55

heavy

1.72

(0.43 to 6.92)

0.43

0.78

(0.60 to 1.01)

0.07

HBP

(reference:   yes)

No

0.58

(0.13 to 2.52)

0.47

1.02

(0.77 to 1.35)

0.88

OA

(reference:   yes)

No

0.10

(0.001 to 10.54)

0.34

0.81

(0.47 to 1.41)

0.47

DM

(reference:   yes)

No

0.42

(0.05 to 3.30)

0.41

0.96

(0.64 to 1.43)

0.85

Height

1.33

(1.21 to 1.46)

0.00001

1.04

(1.01 to 1.06)

0.0001

Weight

1.13

(0.98 to 1.31)

0.08

1.10

(1.07 to 1.13)

0.0001

Waist   circumference

0.88

(0.75 to 1.03)

0.12

0.94

(0.92 to 0.97)

0.0001

Line 309-310 Can the authors change “Thus, our results had an endogeneity bias for racial homogeneity.” to “Thus, our results have an endogenous bias concerning ethnic homogeneity.”

Comment: We changed the sentences as recommended in line 287-292.

Line 310-311 Can this sentence made clearer?

Comment: We changed the sentences as recommended in line 287-292.

Line 314 Can “multi-nation, multi-racial population-based …“ be changed to “more comprehensive, international and multi ethnical studies…”

Comment: We changed the sentences as recommended in line 293-295.

Conclusions

Line 325-326 It’s not clear what the authors mean for “neurological management strategies”. Make this sentence clearer.

Comment: We changed the sentences as recommended in line 297-302.

We would like to thank the reviewers for their insightful comments to help overcome the many shortcomings of our study. We feel confident that these comments would help to upgrade the overall quality of our work. The authors would always be ready to answer to any further comments from the Editors and the reviewers.

Best Regards;

Corresponding author: Jung Soo Lee, M.D., Ph.D.

Round 2

Reviewer 1 Report

The authors have done a nice job of addressing my previous concerns. I hope they consider my additional feedback below for further enhancing their manuscript.

Purpose statement in the abstract and Introduction should include the population studied for specificity (e.g., Koreans aged at least 10 years of age).

Methods: Although this is a secondary data analysis, it should be mentioned where appropriate in the Methods that KNHANES study protocols were approved by an Institutional Review Board (or similar).

Lines 49-50: Again, be careful about the use of “it”. More specificity should be presented to the reader.

Lines 79-81: This sentence is incomplete?

Line 289: Should be “(n=6,577)”

Line 297: Should be “…may affect grip strength”. I would also recommend inserting the population studied here for specificity in a concluding statement.

Line 331: The r-squared abbreviation is no longer needed.

Abstract: Specific statistics are needed in results portion of the abstract. For example, “Hand preference significantly affected grip strength” has no quantitative information to support the statement. Please insert statistics where appropriate.

Author Response

Manuscript ID: ijerph-512144

Title: Normative data on grip strength in a population-based study with adjusting confounding factors: Sixth Korea National Health and Nutrition Examination Survey (2014-2015)

Dear reviewer 1

 The authors are grateful for review to our paper “Normative data on grip strength in a population-based study with adjusting confounding factors: Sixth Korea National Health and Nutrition Examination Survey (2014-2015)”. We went through each single comment with great scrutiny and made sure that we made the appropriate corrections to the manuscript as suggested by the reviewers.

Reviewer #1:

The authors have done a nice job of addressing my previous concerns. I hope they consider my additional feedback below for further enhancing their manuscript.

Purpose statement in the abstract and Introduction should include the population studied for specificity (e.g., Koreans aged at least 10 years of age).

Comment: We corrected the sentences as recommended.

Methods: Although this is a secondary data analysis, it should be mentioned where appropriate in the Methods that KNHANES study protocols were approved by an Institutional Review Board (or similar).

Comment: We added the descriptions in methods as recommended.

‘Informed consents to participate in the study were obtained from all participants by the KCDC. The study was exempt from approval by the Institutional Review Board of Catholic University, College of Medicine, as the study utilized a publicly accessible database, covered by the KCDC.’

Lines 49-50: Again, be careful about the use of “it”. More specificity should be presented to the reader.

Comment: We changed the sentence as recommended (it à the whole nation).

Lines 79-81: This sentence is incomplete?

Comment: We changed the sentence for reducing misunderstanding, as recommended.

Line 289: Should be “(n=6,577)”

Comment: We corrected the sentences as recommended.

Line 297: Should be “…may affect grip strength”. I would also recommend inserting the population studied here for specificity in a concluding statement.

Comment: We corrected the sentences as recommended.

Line 331: The r-squared abbreviation is no longer needed.

Comment: We deleted the r-squared as recommended.

Abstract: Specific statistics are needed in results portion of the abstract. For example, “Hand preference significantly affected grip strength” has no quantitative information to support the statement. Please insert statistics where appropriate.

Comment: we modified the abstract as recommended.

We deeply appreciate reviewers’ time.

Best Regards;

Corresponding author: Jung Soo Lee, M.D., Ph.D.

Reviewer 2 Report

This version of the manuscript has been dramatically improved.  The major strength of the paper is the number of participants across a wide range of ages, occupations, and health related characteristics.  The clinical relevance an novelty of these data are average.  I have not major comments.  The authors have addressed these well in their response to the other reviewers.  There are still a handful of typo and spelling errors.

Author Response

Manuscript ID: ijerph-512144

Title: Normative data on grip strength in a population-based study with adjusting confounding factors: Sixth Korea National Health and Nutrition Examination Survey (2014-2015)

Dear reviewer 2

The authors are grateful for review to our paper “Normative data on grip strength in a population-based study with adjusting confounding factors: Sixth Korea National Health and Nutrition Examination Survey (2014-2015)”. We went through each single comment with great scrutiny and made sure that we made the appropriate corrections to the manuscript as suggested by the reviewers.

Reviewer #2

This version of the manuscript has been dramatically improved. The major strength of the paper is the number of participants across a wide range of ages, occupations, and health related characteristics.  The clinical relevance an novelty of these data are average. I have not major comments. The authors have addressed these well in their response to the other reviewers. There are still a handful of typo and spelling errors.

Comments: We re-checked and corrected the typo and spelling errors as recommended.

The English in this document has been checked by at least two professional editors, both native speakers of English. For a certificate, please see:

http://www.textcheck.com/certificate/T0hAEY

We deeply appreciate reviewers’ time.

Best Regards;

Corresponding author: Jung Soo Lee, M.D., Ph.D.